# Transcendental Idealism of Planner: Evaluating Perception from the Planning Perspective for Autonomous Driving

## Abstract

Evaluating the performance of perception module in autonomous driving is one of the most critical tasks in developing these complex intelligent systems. While module-level unit test methodologies adopted from traditional computer vision tasks are viable to a certain extent, it still remains far less explored to evaluate how changes in a perception module can impact the planning of an autonomous vehicle in a consistent and holistic manner. In this work, we propose a principled framework that provides a coherent and systematic understanding of how perception modules affect the planning of an autonomous vehicle that actually controls the vehicle. Specifically, planning of an autonomous vehicle is formulated as an expected utility maximisation problem, where all input signals from upstream modules jointly provide a world state description, and the planner aims to find the optimal action to execute by finding the solution to maximise the expected utility determined by both the world state and the action. We show that, under some mild conditions, the objective function can be represented as an inner product between the world state description and the utility function in a Hilbert space. This geometric interpretation enables a novel way to analyse the impact of noise in world state estimation on the solution to the problem, and leads to a universal quantitative metric for such purpose. The whole framework resembles the idea of transcendental idealism in the classical philosophy literature, which gives the name to our approach.

## 1 Introduction

Autonomous driving has recently risen as a fast-advancing realm in both industry and academia, and receives a surge of interest from engineering and scientific communities (Yurtsever et al., 2020; Sun et al., 2020). As an intricate system, an autonomous driving vehicle consists of numerous hardware components and interactive onboard modules. As one such core component, the onboard perception module serves as the major source of real-time characterisation of the dynamic environment an autonomous vehicle (AV) navigates through.

To evaluate and improve the perception module, conventional perception tasks (such as detection, segmentation, tracking) have been well defined and corresponding performance measurements are established in computer vision to benchmark performance of perception algorithms (Lin et al., 2014). Despite their great success in driving the development of advanced perceptual information processing modules, almost all such metrics exclusively focus on the perception-level performance in a *deployment-agnostic* fashion, for instance, how close a detected object is to the ground truth, while ignoring the actual impact of the result to the entire AV system. Indeed, not all perception errors translate the same to the planning of an AV. Obviously, miss detecting a vehicle in front of an AV is far more serious than one behind far away. This problem is further compounded by the heterogeneity of perception errors that share little semantics in common ("How dose an error of 5m/s in velocity compare to that of a size 25% larger?"), where intuitive manual engineering is widely used (Caesar et al., 2020). Although these issues are typically addressed by integrating road test in the real world, the process is extremely costly and time-consuming (Wachenfeld and Winner, 2016; Åsljung et al., 2017). In result, tools are in great demand to effectively and efficiently measure the impact of perception to the whole autonomous driving system before deployment on road. Unfortunately, these solutions still remain far less explored in the research literature.

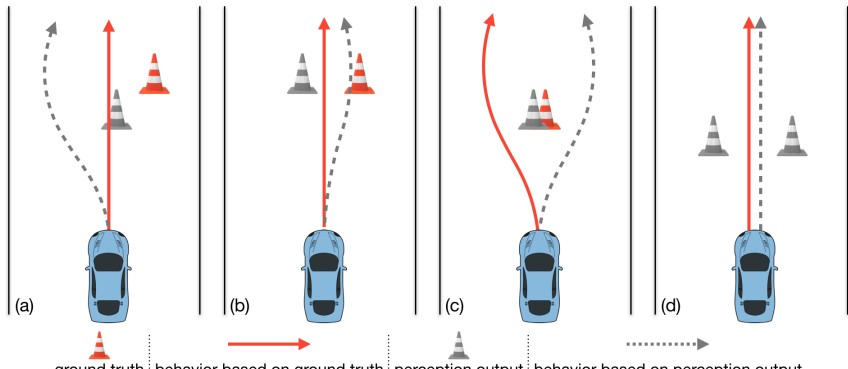

Figure 1: **Illustration of behaviour change v.s. driving cost (best viewed colour).** The change in AV behaviour due to perception error is not always correlated to the cost of consequence. In (a) the AV has to circumvent the erroneously perceived cone by making a large detour. While for (b) the AV only needs to make a slight detour to the right, yet it inevitably hits the cone. In this case, although the behaviour change is far less than that of (a), the consequence is significantly worse ("hitting an object" v.s. "making a large detour"). In (c) the consequence of either way is indifferent to the AV in moving forward, yet the change in behaviour is considerable in terms of spatiotemporal motion. As for (d), if there are two falsely detected cones on both sides, which are close enough to the AV when passing by despite no collision, the AV still decides to maintain the same motion as in the ground truth case. Therefore, the final behaviour of the AV does not change given the perception error, but the cost of passing by two close objects already changes the planning process, which will be missed by the metrics that only look at the AV behaviour or planning result.

Most recently, the community starts to approach this problem with some initial efforts (Sun et al., 2020; Philion et al., 2020; Ivanovic and Pavone, 2021; Deng et al., 2021). Despite some success, these preliminary solutions only exploit certain aspects of the problem, either implicitly relying on weak correlation between behaviour change and driving cost (Philion et al., 2020), inferring the holistic cost via local properties (Ivanovic and Pavone, 2021), or coarse levels (Sun et al., 2020). In this work, we propose a principled and universal framework to quantify how noise in perception input affects the AV planning. This is achieved by explicitly analysing the process of AV planning, in the context of expected utility maximisation (Osborne and Rubinstein, 1994), and evaluating the change of utility values critical to the AV reasoning subject to input perception errors. Under some mild conditions (Section 3.3), we show that this planning process can be formulated as an optimisation problem with linear objective function in a Hilbert space, where utility to optimise is the inner product of an action-wise utility function and the world state distribution represented by perception. This geometric interpretation reveals many natural and insightful properties of the problem, for example, any input error can be decomposed into two parts: one that does not affect the utility comparison (**planning-invariant error**) and the other one that directly changes the planning problem (**planning-critical error**). Based on this novel insight, we derive a metric that quantify how a perception error changes the planning process.

We want to emphasise the necessity of understanding impacts of perception errors on an autonomous driving system via the process of planning, rather than purely from the final result (i.e., the AV behaviour, or the trajectory output from the planning module), as proposed by previous works (Philion et al., 2020). This results from the fact that, the final planning result does not necessarily reflect how AVs evaluate the situation, reason with the environment, and assess the costs of actions. In fact, the correlation between behaviour change and the actual consequence is weak, or even negative in many common cases, as illustrated in Figure 1. Actually, most works implicitly or explicitly integrate some priori knowledge of consequences of perception errors into metric design. The complexity of such impact on autonomous driving, however, is far beyond hand-crafted rules, defeating their purposes despite tremendous amounts of manual efforts, e.g., Deng et al. (2021) assumes that severity of an error should be weighted proportional to the reciprocal of its cubed Manhattan distance to the AV, regardless of its position relative to the AV (in front or behind the AV). In contrast, we make little such presumption and fully rely on the planning process to infer the error consequence in a fully transparent way, which enables our solution to capture many critical cases. In this regard, the core principle of our design resembles the idea in the philosophical system of *transcendental idealism*, proposed by Immanuel Kant in his classical work *Critique of Pure Reason* (Kant, 1998), which argues

that, due to the limitation of the observer's sensibility, cognition of external objects is processed never as they are in themselves, but via the cognitive faculties and subject to the interpretation of the observer's experience. For the same reason, the consequence of environment misrepresentation for an AV due to perception errors is naturally reflected via the change in its planning (the core component of an AV that interprets its environment) and measured by the extra loss incurred, which gives the name to our framework: *transcendental idealism of planner* (TIP).

## 2    LITERATURE REVIEW

**Planning for Autonomous Vehicles.** We consider the behavioural decision and motion planning as the planning process, which generates the vehicle behaviour to execute by the controller given the observation up to the planning time. There is a rich literature to address these fundamental problems in autonomous driving, which can be roughly categorised into canonical module-based and data-driven methods. The former relies on explicit modelling of target accomplishment in optimisation frameworks and seeks the optimal solution as the result (Schwarting et al., 2018). The latter, on the other hand, aims to directly map raw sensor data into the AV behaviour or final vehicle control signals by leveraging the approximation power of deep learning and massive data (Bojarski et al., 2016; Grigorescu et al., 2020), which has attracted increasing attention recently. In this work, we aim to explore the internal mechanism of a planner to gain some insights of the impacts on planning from perception noise, and specifically focus on the module-based planning with explicit target achieving process. Extension of results in this work to data-driven planning is left for future work.

**Behaviour based Metrics for Upstream Modules.** Recent works aimed to assess the performance of perception from the autonomous driving system viewpoint mostly approach the problem in heuristic ways. Considering black-box planning models, Philion et al. (2020) implicitly hypothesise that driving consequences of perception errors are directly correlated to the change in an AV spatiotemporal trajectories planned, and propose the planning KL-divergence (PKL) to measure the impact. While intuitive, it fails to incorporate the context of environment and does not precisely reflect the real cost of input noise in many common traffic scenarios. To deal with the specific problem of object representation, Deng et al. (2021) study how object shapes can affect autonomous driving and devise the support distance error (SDE) to quantify such effect. In a very recent work, Ivanovic and Pavone (2021) start looking into the planning process and employs sensitivity as a probe of input signal's contribution to AV behaviour. This, however, implicitly leverages local-only properties of differentiable cost functions to infer global results. In comparison, our proposed approach captures the big picture of planning process and applies to far more general cases.

## 3    AV PLANNING AS EXPECTED UTILITY MAXIMISATION

To evaluate the performance of perception module from the AV planning perspective in a principled manner, we start by introducing the preliminary basics, and then review the expected utility maximisation (EUM) as the optimal AV action framework. After that, the interpretation of EUM in a Hilbert space is presented, based on which our metric for perception is derived.

### 3.1    PRELIMINARY

We first present the mathematical basics to facilitate the following theoretical analysis. Unless otherwise specified explicitly, all notations follow the standardised one in Goodfellow et al. (2016). A probability space $\{\Phi, \mathcal{F}, \mathcal{P}\}$ is defined by a sample space $\Phi$, an event space $\mathcal{F}$ (a $\sigma$-algebra on $\Phi$), and a Borel probability measure $\mathcal{P}$ on $\mathcal{F}$. A random variable $X : \Phi \to \mathbb{R}^d$ ($d \in \mathbb{N}$) is induced from $\{\Phi, \mathcal{F}, \mathcal{P}\}$ with distribution function $F_X(x)$. When absolutely continuous, $F_X(x) = \int_{-\infty}^{x} f_X(t) \, dt$, where $f_X(x)$ is the probability density function (PDF). $L^2(\mathcal{X}, \rho)$ denotes the space of square-integrable functions, and $\rho$ is a Lebesgue measure accordingly. A Hilbert space $\mathcal{H} = (\mathcal{T}, \langle \cdot, \cdot \rangle)$ is defined on a complete space $\mathcal{T}$ with inner-product $\langle \cdot, \cdot \rangle_{\mathcal{H}}$ and induced norm $\|\cdot\|_{\mathcal{H}}$. Let $S \subset \mathcal{H}$ be a subspace of a Hilbert space $\mathcal{H}$, $S^{\perp} = \{x \in \mathcal{H} | \langle x, y \rangle, \forall y \in S\}$ is the orthogonal complement of $S$ (i.e., the set of all vectors orthogonal to $S$). The linear span of a set $S$ is $\text{span}(S)$. $n_v := v/\|v\|$ is an element of unit length in a normed vector space by normalising element $v$.

### 3.2    AUTONOMOUS VEHICLES AS RATIONAL AGENTS

An AV is an intelligent agent that aims to accomplish some predefined goals in an interactive and uncertain environment. For this, an AV is constantly faced with the problem of planning in the

dynamic environment, and the quality of planning determines how well the goals can be achieved. By the classical EUM theory (Osborne and Rubinstein, 1994), at any given time $t$, an agent aims to achieve the maximum expected reward, defined by the utility function $U$, via execution of the optimal action $a_t^*$ such that

$$a_t^* = \arg\max_{a \in \mathcal{D}_{a,t}} \mathbb{E}\left[U(S_t, a)\right], \tag{1}$$

where $\mathcal{D}_{a,t}$ is the set of all feasible AV actions at time $t$; $s \in \mathcal{S}$, a random variable with distribution function $F_{S_t}(s)$, is the world state at time $t$ in the world state space $\mathcal{S}$; and

$$EU(F_{S_t}, a) \coloneqq \mathbb{E}\left[U(S_t, a)\right] = \int_{s \in \mathcal{S}} U(s, a) \, \mathrm{d}F_{S_t}(s). \tag{2}$$

Intuitively, the utility function encodes the goal or reward the AV is supposed to achieve, for example, to reach a destination in time, to minimise likelihood of collision with other objects, and to avoid sharp change in motion, and $F_{S_t}(s)$ captures uncertainty about the stochastic environment given all prior world knowledge and historical observations up to $t$, which are estimated by modules like localisation and perception. Architectures of many modern AV planners still follow this classical framework (Paden et al., 2016; Buehler et al., 2009; Fan et al., 2018).

### 3.3 EXPECTED UTILITY MAXIMISATION IN THE HILBERT SPACE

To gain some insights into the expected utility of (1) and how input noise is consumed by the planning process, we introduce an interpretation in the Hilbert space to leverage geometric tools available from linear algebra. We first establish the conditions under which a probability measure can be embedded into a Hilbert space, followed by the interpretation of EUM from a geometric perspective in Section 4. For brevity, all proofs are left in Appendix G.

**Theorem 1** (Probability Measure Embeddings in Hilbert Space). *Let $\{\mathcal{X}, \mathrm{d}\}$ be a compact metric space with $\mathrm{d}$ as the metric function, $p$ be a Borel probability measure on $\mathcal{X}$, and $X$ be a random variable on $\mathcal{X}$ with distribution function $F_X(x)$. If $F_X(x)$ is absolutely continuous and the density function $f_X$ is square-integrable, i.e., $f_X \in L^2$, then there exists a unique element[1] $\mu_p \in \mathcal{H}$ such that*

$$\mathbb{E}_X\left[g(x)\right] = \langle \mu_p, g \rangle_{\mathcal{H}}, \ \forall g \in \mathcal{H}, \tag{3}$$

*where element $\mu_p$ denotes the embedding of probability measure $p$ in the Hilbert space $\mathcal{H} = (L^2, \langle \cdot, \cdot \rangle)$, with the inner product given by*

$$\langle g, h \rangle_{\mathcal{H}} \coloneqq \int_x g(x)h(x)\rho(\mathrm{d}x). \tag{4}$$

The critical condition of $F_X(x)$ being absolutely continuous with a square-integrable density function $f_X$ in Theorem 1 is actually general and includes many popular distributions as special cases (see the discussion in Appendix F). Theorem 1 establishes a mapping from probability measures of continuous random variables to $\mathcal{H}$. Additionally, the mapping is also *injective* by the following result.

**Theorem 2** (Injection of Probability Measure Embeddings). *Let $p$ and $q$ be two Borel probability measures defined on a compact metric space $\{\mathcal{X}, \mathrm{d}\}$ with absolutely continuous distribution functions, then $p = q$ almost everywhere if and only if $\mu_p = \mu_q$, where $\mu_p$ and $\mu_q$ are the embeddings of $p$ and $q$ in $\mathcal{H}$, respectively.*

A similar result for mixed distributions is also available in the appendix (Theorem 4). Under the mild conditions in the aforementioned results, the expected utility maximisation of (1) can be rewritten as

$$a^* = \arg\max_{a \in \mathcal{D}_a} \mathbb{E}_{s \sim p(s)}\left[U(s, a)\right] = \arg\max_{a \in \mathcal{D}_a} \langle \mu_p, U_a \rangle_{\mathcal{H}}, \ \forall U(s, a) \in \mathcal{H}. \tag{5}$$

Given the injective correspondence between $p$ and $\mu_p$ established above, we can leverage many tools in algebra (such as inner product, orthogonality, projection, and subspace) to analyse the impact of perception result on AV planning via the EUM in $\mathcal{H}$, denoted "planning utility Hilbert space", where the topological structure is exclusively determined by its inner product.

## 4 PERCEPTION EVALUATION VIA AV PLANNING

In this section, we derive the extra cost of planning incurred by perception errors through the theoretical foundation established in Section 3. While the actual world state characterisation $p(s)$ consists of signals from modules other than perception (e.g., localisation), for brevity we assume that the perception module is the only source for world state estimation in the following discussion.

---

[1]Referred to as a unique class of functions that are equal almost everywhere.

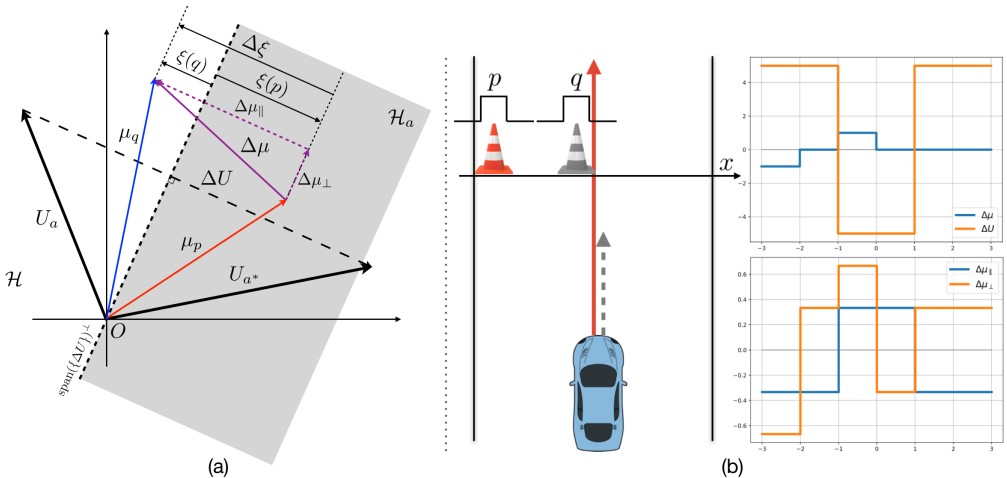

Figure 2: **(a) Illustration of EUM in $\mathcal{H}$.** $\Delta U = U_{a^*} - U_a$ defines the behaviour direction; $\xi$ represents the preference score; $\mu_p$ and $\mu_q$ are the embeddings of the ground truth and noisy perception result, respectively; $\Delta\mu$ is the perception error, which is decomposed into the planning-critical error (PCE) $\Delta\mu_\parallel$, and the planning-invariant error (PIE) $\Delta\mu_\perp$; and the shaded area corresponds to $\mathcal{H}_a$. **(b) A toy example of PCE and PIE (best viewed colour).** The AV is moving forward on a road of width $6m$, where there is a cone in front, the belief of its position is a distribution on a line across the road (the $x$ axis). The ground truth distribution $p$ is $\mathcal{U}_{[-3,-2]}$, a uniform distribution with support $[-3, -2]$, while the perception believes its location (distributed as $q$) is $\mathcal{U}_{[-1,0]}$. The AV has two action options: 1) to move forward ($a^*$, the red arrow), and the utility function is $U_1(x) = -10 \cdot \mathbf{1}_{x \in [-1,1]}$ with $x$ being the position of the cone (only large loss for collision with the cone); 2) to make a hard brake ($a$, the grey arrow), and the utility function is a constant $U_2(x) = -5$ (loss of hard braking is identical regardless of the cone position). In this case, the $\Delta U$ and $\Delta\mu$ are illustrated in the top right, while the decomposition of PIE $\Delta\mu_\perp$ and PCE $\Delta\mu_\parallel$ are on the bottom right. Note that, $\Delta\mu_\parallel$ is of the same shape as $\Delta U$ (up to a negative constant), and $\langle \Delta U, \Delta\mu_\perp \rangle = 0$.

## 4.1 BREAKDOWN OF PERCEPTION ERRORS

Consider a general case where the candidate action set is $\mathcal{D}_a = \{a_i\}$, and each action is associated with a distinct utility function $U(s, a_i) \in \mathcal{H}$ such that

$$\big\| U(s, a_i) - U(s, a_j) \big\|_{\mathcal{H}} > 0 \Leftrightarrow a_i \neq a_j, \; \forall a_i, a_j \in \mathcal{D}_a.$$

Let $a^*$ be the optimal action per EUM of (5) given the ground truth distribution of world state $p(s)$. For a specific $a \neq a^*$, $\Delta U(a^*, a) = U_{a^*} - U_a$, and the planning half-space in $\mathcal{H}$ is

$$\mathcal{H}_a := \{f \,|\, \langle f, \Delta U(a^*, a) \rangle_{\mathcal{H}} > 0, f \in \mathcal{H}\}. \tag{6}$$

Given the actual perception result $q(s)$, $a^*$ is preferred over $a$ by EUM if and only if $\mu_q \in \mathcal{H}_a$, i.e.,

$$\xi(q; a^*, a) := \langle \mu_q, \Delta U(a^*, a) \rangle_{\mathcal{H}} = EU(q, a^*) - EU(q, a) > 0, \tag{7}$$

with $\xi(q; \alpha, \beta)$ denoting the $\alpha$-$\beta$ preference score given $q$, which exclusively decides the result of EUM. As illustrated in Figure 2(a), the final planning is made correctly if and only if

$$\mu_q \in \bigcap_{a \in \mathcal{D}_a / \{a^*\}} \mathcal{H}_a. \tag{8}$$

When there is an error in perception $q(s)$ (i.e., $\big\| \mu_q - \mu_p \big\|_{\mathcal{H}} > 0$), the preference of (7) may be affected, i.e., $\xi(q; a^*, a) \neq \xi(p; a^*, a)$, so is the preference between $a^*$ and $a$ by EUM. To understand how the difference $\Delta\mu = \mu_q - \mu_p$ affects the result of EUM, we further decompose $\Delta\mu$ into two orthogonal components:

$$\Delta\mu = \mu_q - \mu_p = \Delta\mu_\parallel + \Delta\mu_\perp, \tag{9}$$

where

$$\Delta\mu_\parallel = \langle \Delta\mu, n_{\Delta U} \rangle n_{\Delta U} = \frac{\langle \Delta\mu, \Delta U \rangle}{\|\Delta U\|_{\mathcal{H}}^2} \Delta U \tag{10}$$

is the projection of $\Delta\mu$ onto unit vector $n_{\Delta U}$ (denoted *behaviour direction*) parallel to $U_{a^*} - U_a$, and

$$\Delta\mu_\perp \in \text{span}(\{\Delta U\})^\perp \tag{11}$$

---

**Algorithm 1:** TIP Score Evaluation

---

**Input** : An observation sequence from perception $q(\{s_t\}_{t=-\tau}^0)$ and its ground truth $p(\{s_t\}_{t=-\tau}^0)$
**Output** : TIP score of $q(\{s_t\}_{t=-\tau}^0)$

Get the candidate trajectory set $\mathcal{D}_{a,p}$, and the optimal action $a^* \in \mathcal{D}_{a,p}$ from the planner with the ground
  truth perception input $p(\{s_t\}_{t=-\tau}^0)$
Get the candidate trajectory set $\mathcal{D}_{a,q}$ from the planner with the noisy perception input $q(\{s_t\}_{t=-\tau}^0)$
Define the reference action set $\mathcal{D}_a := \mathcal{D}_{a,p} \cup \mathcal{D}_{a,q}$
**foreach** $a \in \mathcal{D}_a$ **do**
$\quad$ Compute four estimates via finite-size samples:
$$\hat{EU}_1 = \frac{1}{n}\sum_{i=1}^n U(s_q^{(i)}, a^*), \hat{EU}_2 = \frac{1}{n}\sum_{i=1}^n U(s_q^{(i)}, a), \hat{EU}_3 = \frac{1}{n}\sum_{i=1}^n U(s_p^{(i)}, a^*), \hat{EU}_4 = \frac{1}{n}\sum_{i=1}^n U(s_p^{(i)}, a)$$
$\quad$ where $\{s_q^{(i)}\}_{i=1}^n$ are $n$ i.i.d. observations from $q(\{s_t\}_{t=-\tau}^0)$, and similar for $\{s_p^{(i)}\}_{i=1}^n$
$\quad$ Compute $\hat{\Delta}\xi(a^*, a; q, p) = \hat{EU}_1 - \hat{EU}_2 - \hat{EU}_3 + \hat{EU}_4$
**end**
Compute and output the result $\mathscr{I}(q, p; U, \mathcal{D}_a) = \min_{a \in \mathcal{D}_a} \hat{\Delta}\xi(a^*, a; q, p)$

---

is the projection of $\Delta\mu$ onto the orthogonal complement of the subspace spanned by the behaviour direction. In the presence of perception error $\Delta\mu$, as illustrated in Figure 2(a), the change in preference score of (7) is only determined by $\Delta\mu_{\parallel}$:

$$\Delta\xi(a^*, a; q, p) = \xi(q; a^*, a) - \xi(p; a^*, a) = \langle \Delta\mu, \Delta U \rangle = \langle \Delta\mu_{\parallel} + \Delta\mu_{\perp}, \Delta U \rangle = \langle \Delta\mu_{\parallel}, \Delta U \rangle. \quad (12)$$

For this reason, we denote $\Delta\mu_{\parallel}$ as the planning-critical error (PCE), and $\Delta\mu_{\perp}$ as the planning-invariant error (PIE). This observation reveals a pivotal fact: *not all errors in world state estimation or perception are of equivalent impact on the AV planning, and the subspace* $\text{span}(\{\Delta U\})^{\perp}$ *contains all errors that would not affect EUM at all.* A toy example of PCE and PIE is shown in Figure 2(b). Given this interpretation, we define the maximum reduction in the preference score of $a^*$ over any candidate actions as the measure of impact from the perception error on AV planning:

$$\mathscr{I}(q, p; U, \mathcal{D}_a) = \min_{a \in \mathcal{D}_a} \Delta\xi(a^*, a; q, p). \quad (13)$$

### 4.2 EVALUATION OF PERCEPTION ERROR IMPACT BY TIP

In practice, combining (7) and (12), the evaluation of perception error impact can be reduced to the calculation of four expected utilities:

$$\Delta\xi(a^*, a; q, p) = \mathbb{E}_{q(s)}\left[U(s, a^*)\right] - \mathbb{E}_{p(s)}\left[U(s, a^*)\right] - \mathbb{E}_{q(s)}\left[U(s, a)\right] + \mathbb{E}_{p(s)}\left[U(s, a)\right]. \quad (14)$$

Computing these expectations in analytical forms typically requires strong assumptions on the forms of both utility and distribution functions, which substantially limits the flexibility and representation capacity. To allow for maximum representation flexibility, we resort to numerical methods and estimate the expected utilities from finite-size samples of world states, and show that the solution is both statistically consistent and uniformly efficient under the mild conditions in Theorem 3. Specifically, for a fixed action $a$, given an i.i.d. sample of the utilities $\{U(S_i, a)\}_{i=1}^n$ with $S_i$ drawn from the state distribution $p_S(s)$, an unbiased estimator of the expected utility based on U-statistics is

$$EU_a = \frac{1}{n}\sum_{i=1}^n U(S_i, a). \quad (15)$$

Under many common practical conditions, we show that fast convergence rate via uniform bound of the estimator of (15) can be achieved by the following observation.

**Theorem 3** (Exponential Convergence Rate). *If there exists an $M \in \mathbb{R}$ such that $U(S, a) < M$ almost surely, then*

$$\Pr\left(\left|EU_a - \mathbb{E}\left[U(S, a)\right]\right| > \varepsilon\right) < 2e^{-\frac{n\varepsilon^2}{2L}}, \; \forall \varepsilon > 0, \; L = \min\left(M^2, \text{Var}(U(S, a)) + M\varepsilon/3\right). \quad (16)$$

The exponential convergence rate at $O(e^{-n})$ provided by Theorem 3 is significant in the sense that it depends on (i) neither the dimensionality of the original state space $\mathcal{S}$ (i.e., the curse of dimensionality is not invoked), nor (ii) the distribution $p_S(s)$ and utility functions (i.e., $U(S, a)$ and $p_S(s)$ can take any arbitrary forms). To facilitate the understanding of our approach, the pseudo code for evaluating TIP is provided in Algorithm 1, which sketches the basic routine to compute TIP score of a perception input sequence $q(\{s_t\}_{t=-\tau}^0)$ from $t = -\tau$ to $t = 0$ for planning at $t = 0$.

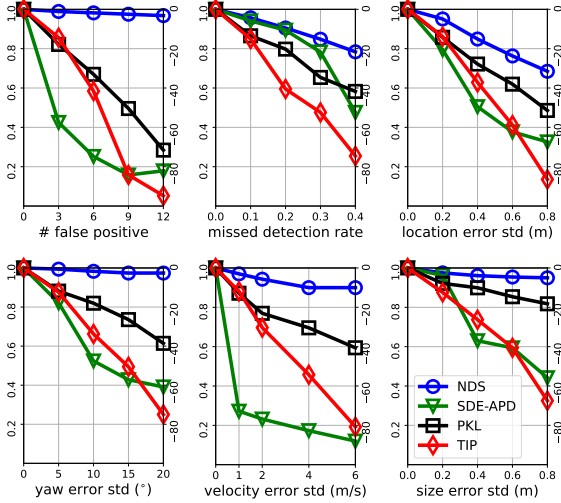

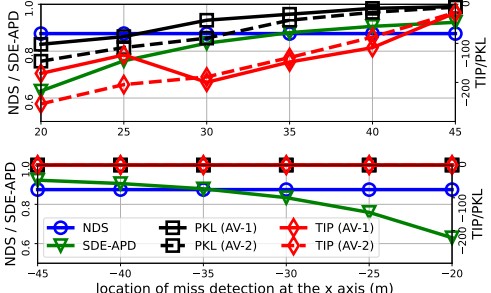

Figure 4: **Metrics for AVs of different driving styles.** A miss detected stationary obstacle is placed in front of an AV at $x$=0 moving at 14m/s along the $+x$ direction. AV-1 ("jerk-averse") is optimised for driving comfort with braking capped at -4m/s$^2$, while AV-2 ("collision-averse") is for safety and can brake as much as -6m/s$^2$. Both SDE-APD and PKL consider closer miss detection is worse than further ones. TIP, however, reveals that AV-1 regards the one at 30m the worst, since the collision is inevitable even if the obstacles at 20m and 25m are detected; yet miss detection at 30m leads to a collision that could have been avoided otherwise. In contrast, no other metrics provide this fine resolution at this level. When the miss detection happens behind the AV, both TIP and PKL ignore its impact unlike the other two. Note the symmetry of NDS and SED-APD in both directions of the x-axis.

Figure 3: **Comparison of metrics on different cases of synthetic noise.** The left vertical axes are for NDS and SDE-APD, and the right are for PKL and TIP. Note that, NDS saturates in several cases once the noise reaches some levels, and sensitivity of TIP to various types and levels of noise is generally more consistent than the other three (SDE-APD is computed by SDE-APD@$t$=1s for velocity noise).

## 5 EMPIRICAL STUDY

In this section, we evaluate how the proposed TIP works in practice via extensive experiments on both synthetic and real data. Due to the space constraint, more details are left in the appendix.

### 5.1 BASIC SETTINGS

All AVs used in the experiment are based on the same type of regular passenger vehicles. The planner deployed in the autonomous driving system is derived from the popular open-sourced project of Apollo (Fan et al., 2018). It consists of various sub-modules of routing, object motion prediction, cost generation, path finder, and trajectory optimisation. At each planning time instant, these sub-modules (except the trajectory optimisation) analyse the environment, input history, and establish the target utility function for final trajectory optimisation, i.e., the objective utility function $U(\cdot, s)$ is first created with the perception input as (part of) hyper-parameters. The path finder then provides multiple initial paths as candidates for path-wise trajectory optimisation, and the final choice is determined by a utility decider. The goals the planner strives to achieve include motion smoothness, traffic rule compliance, safety, progress to the destination, *etc*. The planner has been extensively verified via rigorous road tests in major cities with millions of population (See Appendix B for more details).

All experiments are implemented in scenarios as the standard protocol in autonomous driving (Riedmaier et al., 2020). Scenarios used are collected from real world road tests (see more details in Appendix D). We consider the planning problem at a particular frame in a scenario at a time, and evaluate the utility of an action (a spatiotemporal trajectory the AV executes) for the next three seconds, following the basic setup of (Philion et al., 2020). For comparison, three baselines are adopted from the spectrum of perception metrics: 1) at the conventional end, nuScenes dataset score (NDS) (Caesar et al., 2020) combines several traditional scoring results for 3D object detection into a single performance measure, 2) SDE average precision distance-weighted (SDE-APD) (Deng et al., 2021) focuses more on support distance errors near the AV in an ego-centric fashion, and 3) PKL (Philion et al., 2020) serves as the representative for AV behaviour-driven metrics.

### 5.2 RESULTS ON SYNTHETIC DATA

In the first set of experiments, we aim to gain some understanding of various metrics in reaction to common types of perception noises. A dataset is synthesised from our curated road test scenarios by adding controlled noise to the 3D object ground truth of vehicles, to enable clear observation on

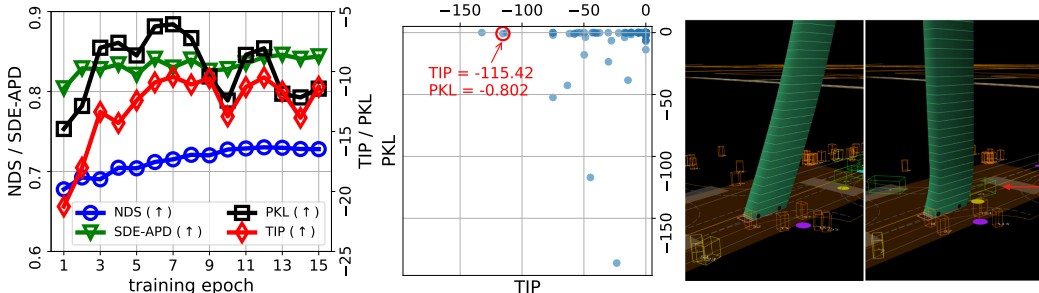

Figure 5: **Comparison of metrics on real data (best viewed colour).** Left: Metrics on different checkpoints during training. Middle: Scatter plot of impacts of perception noise measured by TIP and PKL. Note the number of data points close to the $x$-axis (PKL = 0), which correspond to critical errors in planning due to the perception noise captured by TIP yet missed by PKL since the AV behaviours are similar with and without the noi se. Right: First one is the ground truth with corresponding AV behaviour (the spatio-temporal trajectory represented by the green tube with the $z$-axis as the temporal dimension); second one shows that a false positive (pointed to by the red arrow) causes an outrageous planning error with a jerk of $-76.4$m/s$^3$ while the typical limit is around $-1.0$m/s$^3$ (Wang et al., 2018), despite a mild change in behaviour by PKL. PKL and TIP of this case are highlighted by the red circle in the scatter plot (the middle figure). See details in Figure 9 and Figure 10 of Appendix C.1.

the sensitivity of metrics to specific types of perception errors. For this, 1000 5s-long scenarios are assembled, with the number of objects per scene between 30 and 500, where AVs are moving on an average speed at least 5m/s. The ground truth is annotated by professionally trained human operators. All objects in the scenario are labeled with location, heading, category, and bounding box from 3D point clouds recorded from LiDARs on AVs during road tests.

In total, six types of errors are considered. The false positives are tested by adding "ghost" vehicles scattered within a 70m-by-30m box centred at the AV, with motion properties randomly perturbed from the AV. The miss detection is created by removing objects from the ground truth randomly with a certain probability (i.e., miss detection rate). Other noises involving location, yaw, velocity and size are sampled from zero-mean Gaussian distributions with different variances and added to corresponding properties of objects in ground truth. The comparisons are shown in Figure 3. While all metrics are negatively correlated with all six types of perception noises, NDS saturates in some types (e.g., velocity) due to its design. SDE-APD, also involving manual engineering, exhibits varying sensitivity at different noise levels (especially for the velocity, as the default matching threshold 0.2m is easily overwhelmed by speed noise larger than 1m/s. Selectivity of TIP tends to be more consistent than PKL, in the sense that, while both may predict results at similar dynamic ranges when the noise is mild, the former indicates larger loss when input perception errors intensify in most cases.

We further investigate behaviour of TIP in some individual cases. In a typical miss detection scenario, we remove a stationary vehicle in front of or behind moving AVs. As shown in Figure 4, outcomes rendered by TIP change with different planner settings, and it predicts the miss detection of the worst loss at the border of collision events when the accdident could have been avoided by a small margin if the obstacle is successfully detected. This reveals the superior resolution of TIP in identifying critical events from the planning perspective that would have been missed by other baselines. The case also shows that NDS and SDE-APD fail to distinguish errors at both sides of the AV, due to their spatial or directional homogeneity by design.

## 5.3 RESULTS ON REAL DATA

In the second set of experiments, we study the results from the real perception module deployed on our AVs, which is exemplified by a 3D object detection model that predicts class, location, heading, velocity and size of objects from LiDAR point clouds. TIP is independent of the specific detector and can be applied to various methods (Lang et al., 2019; Yin et al., 2021; Shi et al., 2020). We adopt an end-to-end multi-view fusion (MVF) based model to synergise the birds-eye and perspective views of point clouds (Zhou et al., 2019). The model is trained on 780K LiDAR sweeps using annotations of vehicle, pedestrian and cyclist with detection range [-67.2m, 124.8m]×[-51.2m, 51.2m].

A typical challenge in developing the perception model is to determine how much training is needed to reach a satisfactory level of performance. Conventional solutions require a variety of heterogeneous

Table 1: Comparison of different detectors across metrics (↑).

| Detector | NDS | SDE-APD | PKL | TIP |
|---|---|---|---|---|
| MVF + Tracking | 0.730 | 0.843 | -8.1 | -10.5 |
| MVF (PN-T) | 0.693 | 0.852 | -9.2 | -11.7 |
| 5F-MVF (PN-T) | 0.744 | 0.878 | -7.9 | -9.1 |

Table 2: Subjective evaluation.

| Metric | v.s. TIP |
|---|---|
| NDS | 17.6% / 82.4% |
| SDE-APD | 34.5% / 65.5% |
| PKL | 38.8% / 61.2% |

metrics to measure different aspects of the algorithm, including mean average precision (mAP) for detection and mean squared errors in predicted motion properties. Recently, some unified metrics like NDS (Caesar et al., 2020) are also proposed by manual engineering. These metrics hardly predict outcome of the driving quality improvement of a perception model change, and in most cases the conclusion can only be made from large-scale real road tests, which is almost infeasible for such purpose (Wachenfeld and Winner, 2016; Åsljung et al., 2017).

We evaluate the performance of our 3D object detection model on the same benchmark as in Section 5.2 and compare the model output against the ground truth. The model is trained for 15 epochs, and results are reported in the left of Figure 5. Not surprisingly, NDS tends to increase as the model training progresses and the final checkpoint models usually achieve the best performance since NDS combines the errors that are aligned with the loss functions optimised during training. When evaluated with the AV involved, however, the observation is not quite similar. SDE-APD implies that the training, without the AV context, seems to struggle with improving results on close-by objects as the losses are dominated by large number of far-away yet more challenging objects. From either behaviour or planning perspectives, both TIP and PKL indicate that the last checkpoint model is not among the best possible models during training. Instead, models somewhere in the middle of the training can provide better autonomous driving performance. Actually, neither TIP nor PKL is improved significantly beyond the 7th epoch, suggesting that early termination of training may be even more beneficial to driving quality. More importantly, we notice that TIP disagree with PKL on scenarios across models of top performance, where there are quite some critical cases identified by TIP yet missed by PKL. The difference is illustrated in the middle of Figure 5 by the scatter plot of randomly sampled scenarios, where the PKL values are almost zero while the TIP scores are non-trivial for quite a number of cases, suggesting the drastic impact of the perception errors on the AV planning process despite similar AV behaviour outputs with or without these errors (related individual examples are discussed in Appendix C.1).

To compare other 3D detectors for offline applications (e.g., auto labelling (Qi et al., 2021)), we implement two offline models with PillarNet (Shi et al., 2022) enhanced with transformer modules as the basic detector. The first one, denoted MVF (PN-T), uses the point cloud only from the current frame for prediction. The second one, denoted 5F-MVF (PN-T), leverages 5 consecutive frames around the current one to predict. Results are reported in Table 1. Both offline models, with far less restriction on resources, have better performance by SDE-APD. MVF (PN-T), however, cannot produce precise velocity out of observation from only one frame, which leads to inferior performance by other three metrics (MVF + tracking is the onboard model discussed above). 5F-MVF (PN-T) delivers overall best results across all metrics, despite the marginal gap computed by PKL.

To further justify the soundness of the proposed approach on the scenario level, we also implement a set of subjective evaluation similar to that in (Philion et al., 2020). We collect 258 pairs of scenarios with actual perception noises and check weather TIP, PKL or NDS disagree on the relative severity (i.e., one believes the perception error in scenario A is worse than that in scenario B while the other one thinks alternatively). These scenario pairs are compared and rated by 10 randomly selected human drivers to decide on which is worse from the human perspective. The result reported in Table 2 suggests that human drivers side with TIP over other three baselines.

## 6 CONCLUSION

In this work, we propose a principled framework to evaluate perception from the perspective of planning for autonomous driving. Our approach explicitly exploits the properties of module-based planners and effectively identifies perception noises that may cause large planning change in the context of expected utility maximisation. Extensive experiments on both synthetic and real data confirm that our approach is capable of distinguishing perception errors that would not be separated by conventional metrics or those only exclusively focusing on AV behaviours.

## 7 ETHICS STATEMENT

Autonomous driving directly involves interaction with human beings, animals and other assets in the real world. Any performance measure tools for autonomous driving may not identify or cover all possible failure cases of AVs that may lead to negative or even catastrophic consequences. TIP is not an exception, despite the principle to precisely reflect the potential loss in planning by design. It is critical that any researchers and practitioners of TIP should still implement standard and comprehensive safety protocols to ensure legitimate compliance to appropriate laws and rules, to minimise the likelihood of any potential negative impacts.

## 8 REPRODUCIBILITY STATEMENT

The submission includes three major technical parts: theory, implementation, and empirical study. The main results of theoretical analysis and treatments are presented in Section 3.3 and Section 4.2. The detailed explanation and complete proofs are presented in Appendix F and Appendix G. The computation of TIP is provided as pseudo-code as in Algorithm 1. For the empirical study, the planner is derived from the open-sourced Apollo with configuration fine-tuned on real road test data as depicted in Appendix B. NDS is computed via the open-sourced implementation. SDE-APD is implemented in house according to the original paper (Deng et al., 2021). PKL is implemented following the open-sourced project.

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

# Appendix

## Table of Contents

## A ADDITIONAL LITERATURE REVIEW

**Expected Utility Maximisation**. Almost all solutions proposed so far in the artificial intelligence realm to develop intelligent agents have followed the rational-action strategy as opposed to the human-like behaviour strategy, i.e., the optimal decision made by a rational agent should optimise some achievement measurement on consequences of its decision, which is, ideally, well aligned with the predefined high-level goals (Russell and Norvig, 2009). Among many possible formulations of the outcome measurement, the *expected utility* (EU) hypothesis is one of the most popular frameworks in the decision theory (von Neumann and Morgenstern, 1944). While first introduced to characterise human behaviours in microeconomics, the idea also finds great success in many other domains, including the optimal decision for artificial intelligence systems (Osborne and Rubinstein, 1994).

**Embedding Probability Measures in Hilbert Space.** Embedding probability measures into a Hilbert space has been explored in the literature for kernel methods (Berlinet and Thomas-Agnan, 2011; Smola et al., 2007). These methodologies exclusively focus on the reproducing kernel Hilbert space, which is spanned by Mercer kernel functions. However, the implicit requirement on function continuity may restrict its applicability to our problem, where either utility and probability density functions may be discontinuous. In this work, we consider a less restrictive Hilbert space where only square-integrability is needed and more flexible functions are possible.

## B AUTONOMOUS VEHICLE PLANNER

We start by introducing the autonomous vehicle planner, which provides the fundamental toolkit for the proposed evaluation framework. Our planner is designed to control an Level 4 AV running in urban areas of major modern cities. Its modulized architecture is similar to many popular utility-based planners (e.g., (Fan et al., 2018)), which consists of four major components as illustrated in Figure 6:

- The *predictor* infers the motion information $s_m$ in the future (i.e., $t > 0$) for all dynamic road objects from perception input history (i.e., $t \leqslant 0$) up to the planning time (i.e., $t = 0$).

- The *action proposer* analyses the current environment at the planning time from (1) the perception input, (2) future object motion input, and (3) other inputs (e.g., localisation, traffic lights, semantic maps, routing path, etc.), and proposes various sets of behaviours (e.g., "go straight" and "lane change") for the AV with an initial feasible spatiotemporal trajectory for each set.

- The *trajectory optimiser* takes results of above components as input, and finds the optimal spatiotemporal trajectory for each behaviour set by numerically solving an optimisation problem with the initial feasible trajectory from the proposer as the starting point.

- The optimal trajectories from all behaviour sets are then submitted to the *action decider*, which assemblies all information to evaluate the utilities of different candidate actions (with corresponding optimal spatiotemporal trajectories), and makes the final decision on the $a^*$.

Similar to many popular architectures (e.g., (Fan et al., 2018)), our planner utility function $U(a, s)$ is of the general form

$$U(s, a) = \sum_i \lambda_i U_i(s, a) + U_s(s) + U_a(a),$$

where $\lambda_i$ are the (static) coefficients, the atomic element function $U_i$ depending on both a and s characterises the "compatibility" of action $a$ and scenario $s$, $U_s(s)$ depicts the current environment, and $U_a(a)$ evaluates the quality of the action. These terms can be categorized into the following groups.

- The *smooth motion* group encourages motion without abrupt change in acceleration, and penalises large jerks (i.e., the derivative of acceleration).

- The *safety distance to road obstacles* group is designed to keep the AV away from other road objects to minimise the collision likelihoods. This distance is defined as $\ell^2$ distance between the AV spatiotemporal sweeping contour and a foreign object on the road.

- The *legal motion satisfaction* group is designed to enforce the AV to strictly follow all applicable traffic rules when in motion. For instance, the cost for crossing solid yellow lines

is so significant as that such behaviour is prohibited unless a collision cannot be avoided otherwise. Some other legal options also come at certain prices too to discourage high-risk behaviour (e.g., lane changes in crowded scenes).

- The *progress to the destination* group aims to guide the AV to achieve the goal in the big picture and reach the final destination.

The aforementioned planner deployed onboard our AVs have gone through rigorous road test in urban areas of major cities with millions of population. Results of more than 10,000 miles on average tested every week indicate that the planner achieves 111.3 miles per intervention (MPI), which suggests that the planner used in this work is a reasonable and validated one.

## C MORE COMPARISONS TO RELATED METRICS

In comparison to the other advanced metrics (PKL (Philion et al., 2020), IPA (Ivanovic and Pavone, 2021), SDE (Deng et al., 2021)) that are recently proposed for more effective perception evaluation, our approach provides a universal and principled solution to evaluate the impact of perception noise from the perspective of the planning process of an AV.

### C.1 COMPARISON WITH PKL

Here we provide more empirical results to better understand the difference between the proposed TIP and PKL (planning KL-divergence) (Philion et al., 2020).

**Results on Synthetic Data**

Figure 7 demonstrates a scatter plot for scene-wise TIP and PKL results on the synthetic data generated as described in Section 5.2 with 6 false positives per scene. It is observed that some results are very close to either $x$- or $y$-axis (the top right corner), suggesting that TIP and PKL deviate in determining if a perception error (i.e., false positive) is crucial to planning on these cases. A typical scenario of such disagreement is shown in Figure 8, where the behaviour of the AV does not change significantly with ground-truth or noisy perception inputs (PKL = -0.248), yet the planning process has changed quite a lot (TIP = -61.654) due to the affinity of false positive objects that has drastically change the planning cost to close objects. In this case, TIP is capable of detecting serious perception errors that PKL fails to identify.

**Results on Real Data**

On the real data, we also have similar observations, and demonstrate the actual scene for one such scenario in Figure 9. As shown in this case, a false detection of a vehicle in front of the AV does not change the behaviour considerably (PKL = -0.802), while the significant planning cost change is reflected by TIP with value -115.42. More individual examples are shown in Figure 10

Overall, on both synthetic and real data, we show that the proposed TIP can efficiently and effectively capture perception errors critical to the planning that may be missed by PKL. This confirms our motivation to exploit the actual AV planning process, as opposed to the AV behaviour (i.e., the result of planning), to gain insights into the impact of input perception error on the whole AV system.

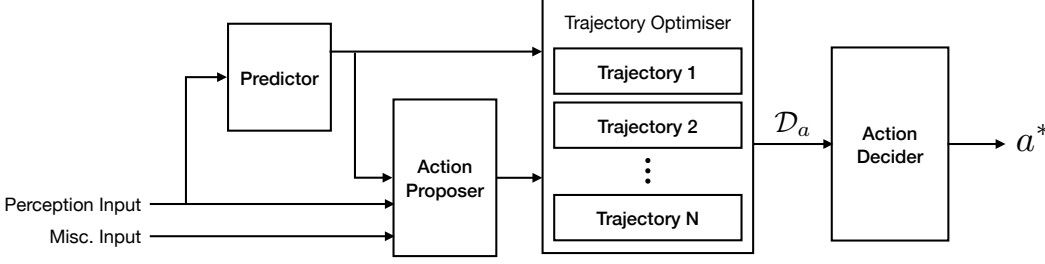

Figure 6: Diagram of the major components in our planner which is used for computing the proposed perception evaluation metric TIP.

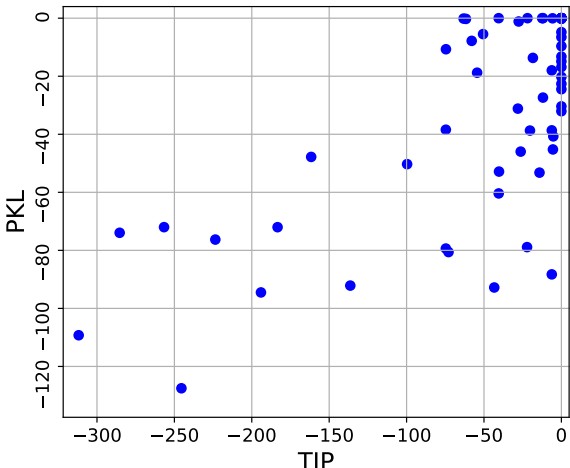

Figure 7: **Result on the false positive synthetic data.** The data points (downsampled for clarity) close to $x$-axis (PKL = 0) correspond to the cases where the AV behaviours under ground truth and noisy perception inputs are identical. The data points close to $y$-axis (TIP = 0) correspond to the cases where the AV planning preference between the optimal action and others are identical under ground truth and noisy perception inputs. Note the number of cases where TIP disagrees with PKL on the impact on AV planning.

## C.2 COMPARISON WITH IPA

IPA (injecting planning-awareness) is recently developed in (Ivanovic and Pavone, 2021) to encode the planning error based on the hypothesis that the impact of object location error is proportional to the gradient magnitude of the planning cost functions involving the AV-object distance. This solution however requires differentiability of the planning cost functions, while our approach does not and is thus more applicable to the Level 4 AVs that are typically structured as a modularized pipeline of individual components including perception, prediction, planning, etc. Even more serious is that it fails to account for all cases since the local properties (gradients) do not always reflect the global ones (overall losses). To illustrate this, consider a scenario, where the cost of AV being close to an object is $1/d$. Now assume that there are the following two cases of object location errors.

- Case one: The ground truth distance of an object to the AV is 1m, and the noisy distance estimated by perception is 0.9m. Per the metric IPA defined in (Ivanovic and Pavone, 2021), the result is

$$\left|\frac{\mathrm{d}}{\mathrm{d}d}(1/d)\Big|_{d=1}\right||\Delta d| = 1 \times |1.0 - 0.9| = 0.1,$$

while the actual cost difference is $\left|\frac{1}{0.9} - \frac{1}{1}\right| = 1/0.9 - 1 = 0.111$.

- Case two: The ground truth distance of an object to the AV is 2m, and the noisy distance estimated by perception is 2.5m. Per the metric IPA defined in (Ivanovic and Pavone, 2021), the result is

$$\left|\frac{\mathrm{d}}{\mathrm{d}d}(1/d)\Big|_{d=2}\right||\Delta d| = 0.25 \times |2.5 - 2.0| = 0.125,$$

while the actual cost difference is $\left|\frac{1}{2} - \frac{1}{2.5}\right| = 0.5 - 1/2.5 = 0.1$.

Obviously, the metric IPA of case two is larger than case one, while the actual error in planning cost is the other way, as Taylor series up to first order terms adopted by IPA cannot precisely delineate the cost function value change over large range input variation.

Figure 8: **Illustration of AV behaviours in ground truth and synthetic scenes with false positives.** The green tube represents the spatiotemporal trajectory of the AV with the z-axis as the temporal dimension (same for the rest). Bold solid lines are boundary of driving areas (e.g., curbs, vegetarian zoom dividers), while light solid lines are centre lines of vehicle lanes with dashed lines as the lane boundaries. Road objects are marked with 3D bounding boxes in green. Sub-figures in the first (second) row are birds-eye view (side view) of the scene, and sub-figures in the left (right) column correspond to ground truth (noisy) perception input (same for the rest). In this case, the AV intends to move forward under the ground truth perception input (the left column); in the presence of perception input noise (the right column), the AV behaviour remains almost unchanged (PKL = -0.248), since two false positive vehicles (pointed by red arrows) on its both sides force the AV to keep moving straight, yet the close-to-object cost (safety distance to road obstacles) has changed considerably during planning, and is reflected by the score of TIP -61.654.

## D  SCENARIO COLLECTION

The scenarios used in this work are curated from AV road test in real world from public roads in urban areas of megacities, e.g., central business districts, populated residential communities, major commercial areas, etc. Each scenario is a 10s-long excerpt extracted from a continuous interval of road test, which consists of 1) all raw data recordings (LiDAR point clouds, camera images, positioning signals, etc.) from the road test during the interval, and 2) the portion of offline generated high-definition (HD) and birds-eye view (BEV) raster maps that cover the field of perception during the interval. The duration of road test ranges from tens of minutes to several hours, and covers various times of both weekdays and weekends from early morning till late night during a period of more than one year, providing a rich blending in weather condition (e.g., sunny, cloudy, rainy, and snowy), traffic intensity (e.g., rush hours on highways and crowded streets during holidays), road participant diversity (e.g., private cars, cyclists, pedestrians, and emergency vehicles), and so forth. The scenarios are selected from non-trivial situations (i.e., those with few traffic participants are filtered out) with balance in AV motion speed, diversity of traffic participants, weather, geographical locations, etc.

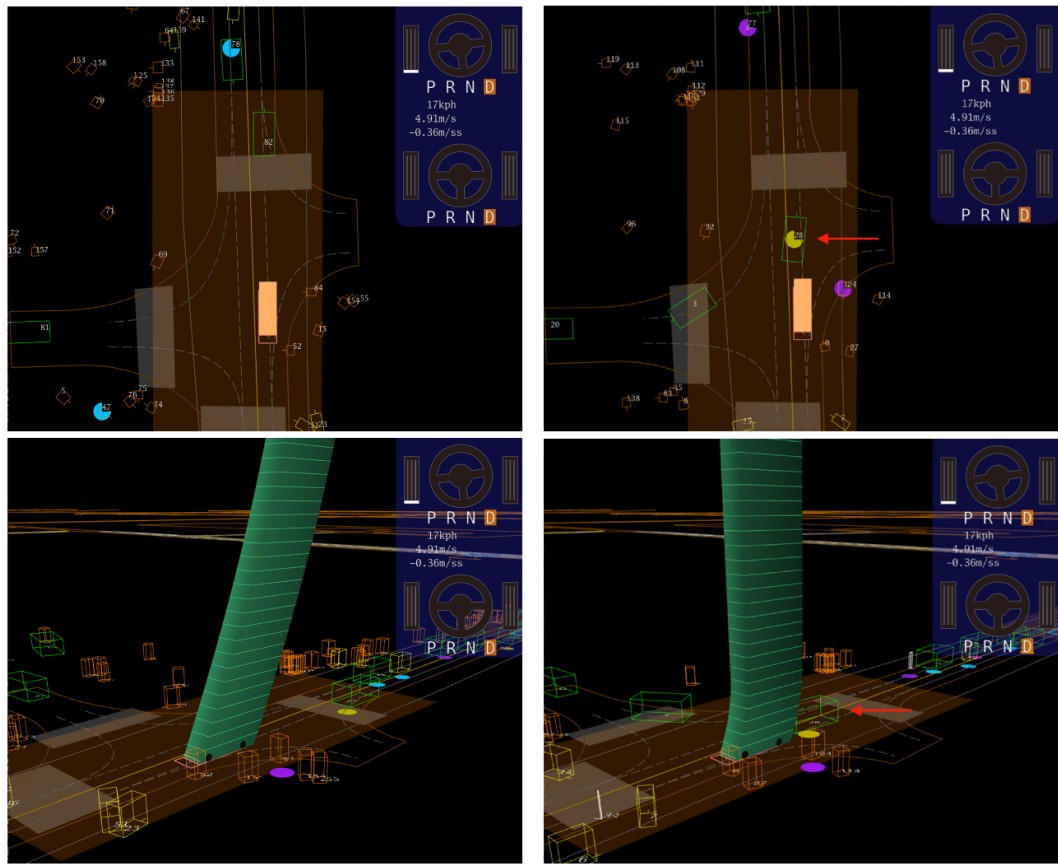

Figure 9: **Illustration of AV behaviours in reaction to ground truth and actual noisy perception inputs.** Under the ground truth perception input (the left column), the AV is clear to move forward with a soft braking to keep distance to another vehicle ("82") in front. Given the noisy perception input (the right column), however, the AV has to hard brake to avoid potential collision with the false positive vehicle close to it in front (marked by the red arrow). In either cases, since the AV speed is slow and is braking (either soft or hard), the difference in behaviour is insignificant (PKL = -0.802), yet the consequence of the false positive is by no means trivial: the false positive causes a hard brake and virtual collision (between the behaviours under ground truth perception input and false positive), which is precisely captured by the proposed TIP (TIP = -115.42). The kinematic motion for the ground truth scenario (bottom left) is $a = -0.36\text{m/s}^2$, $j = -0.72\text{m/s}^3$, and for the noisy scenario (bottom right) is $a = -0.36\text{m/s}^2$, $j = -76.4\text{m/s}^3$. **Note how sharp the braking changes in presence of the noisy perception** (jerk: $-0.72\text{m/s}^3$ v.s. $-76.4\text{m/s}^3$). Clearly, this is a critical error from the system perspective.

## E    BREAKDOWN OF TIP

In order to facilitate understanding of the TIP evaluation process, we have illustrated the evaluation steps in Figure 11, where a typical false negative case is used for the analysis.

## F    EXAMPLES AND NON-EXAMPLES OF SQUARE-INTEGRABLE DENSITY FUNCTIONS

Theorem 1 in the main text requires square-integrability of a density function,which includes many popular cases that may be used for constructing the utility function for planning.

**Example 1** (Bounded PDFs). *If both the support and range of the PDF $f(x)$ of an random variable is bounded, then $f(x)$ is square-integrable, e.g., uniform distributions.* ∎

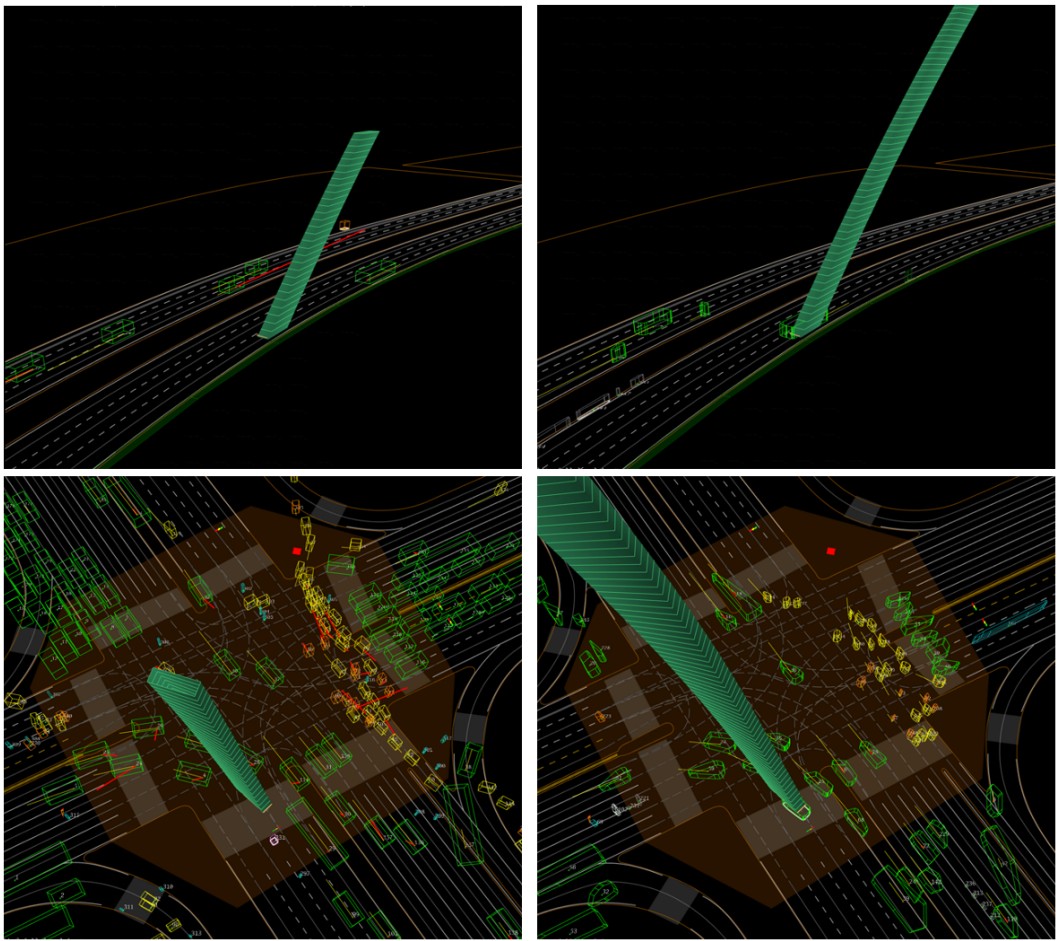

Figure 10: **Illustration of more AV behaviours in reaction to ground truth and actual noisy perception inputs.** Two more outrageous perception errors are shown where an object location is improperly perceived such that it is superimposed with the AV. The ground truth is shown on the left and the actual noisy perception is on the right. For the top case: TIP = -132.4, PKL = 0.0, acceleration $a = 0.03\text{m/s}^2$, jerk $j = 0.09\text{m/s}^3$ (for both ground truth and actual noisy perception scenarios). For the bottom case: TIP = -75.0, PKL = 0.04, $a = -0.35\text{m/s}^2$, $j = -2.43\text{m/s}^3$ (for the ground truth scenario), and $a = -0.35\text{m/s}^2$, $j = -2.61\text{m/s}^3$ (for the actual noisy perception scenario).

**Example 2** (Parametric PDFs). *PDFs of many popular parametric statistical models are square-integrable, e.g., (sub-)Gaussian, (sub-)Laplace, Gamma (including exponential, Erlang, and $\chi^2$ distributions), etc.* ∎

**Example 3** (Mixture Models of Countable Components with Square-Integrable PDFs). *The PDF of a mixture model is of the form:*

$$f(x) = \sum_i \alpha_i f_i(x), \ \alpha_i > 0, \ \sum_i \alpha_i = 1, \tag{17}$$

*where $f_i(x)$ is the PDF of the $i$-th component out of the countable set $\{f_i(x)\}$. $f(x)$ of (17) is square-integrable if $\forall i, f_i \in L^2$ and $M = \sup_i \|f_i\|_{\mathcal{H}} < +\infty$ as*

$$\int |f(x)|^2 \,\mathrm{d}x = \int \sum_{i,j} \alpha_i \alpha_j f_i(x) f_j(x) \,\mathrm{d}x = \sum_{i,j} \alpha_i \alpha_j \langle f_i, f_j \rangle_{\mathcal{H}} \tag{18}$$

$$\leqslant \sum_{i,j} \alpha_i \alpha_j \|f_i\|_{\mathcal{H}} \|f_j\|_{\mathcal{H}} \leqslant M^2 < +\infty.$$

*A variety of mixture models are included such as Gaussian mixture models and mixtures of Gamma distributions.* ∎

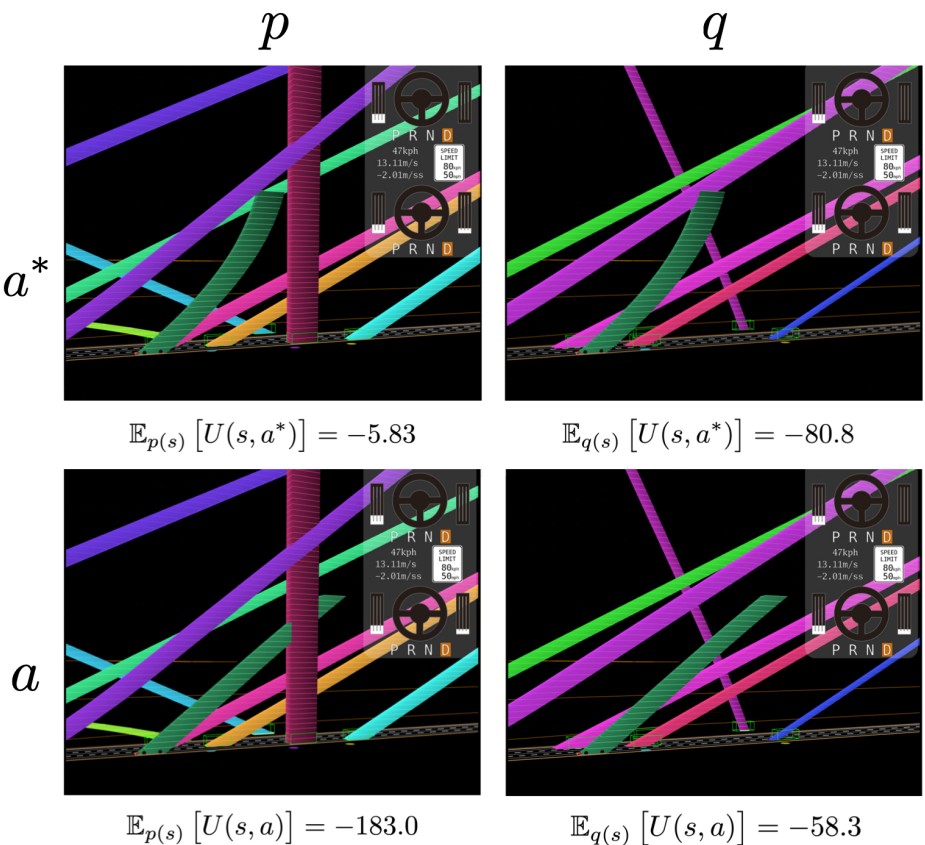

$$\mathbb{E}_{p(s)}\big[U(s,a^*)\big] = -5.83 \qquad\qquad \mathbb{E}_{q(s)}\big[U(s,a^*)\big] = -80.8$$

$$\mathbb{E}_{p(s)}\big[U(s,a)\big] = -183.0 \qquad\qquad \mathbb{E}_{q(s)}\big[U(s,a)\big] = -58.3$$

Figure 11: **Breakdown of TIP for a typical miss detection scenario.** A stationary vehicle $30m$ in front of the AV is missed by the onboard object detector. Four scenes are: 1) (top left) ground truth behaviour ($a^*$) in ground truth environment ($p$) with EU -5.83 (hard brake to avoid a collision); 2) (top right) ground truth behaviour ($a^*$) in noisy perceived environment ($q$) with EU -80.8 (hard brake without any obstacles in front); 3) (bottom left) candidate behaviour ($a$) in ground truth environment ($p$) with EU -183.0 (move forward and collide with the object in front); 4) (bottom right) candidate behaviour ($a$) in noisy perceived environment ($q$) with EU -58.3 (move forward). Planned (for AV) or predicted object motion in a scene is rendered as a coloured spatiotemporal trajectory (3D tube with z-axis as time), e.g., each tube consists of locations of the corresponding object at time $t$ ($t$ is the coordinate of the z- or vertical axis). Note that in the lower left scene the candidate AV behaviour $a$ (move forward at a constant speed) is evaluated against the ground truth environment ($p$), which collides with the object at the end of $3s$. Corresponding utilities are shown under all cases, and TIP in this case is $-199.7$ according to the calculation of expected utilities in (15).

On the other hand, since $\ell^1$ and $\ell^2$ norms are not necessarily equivalent in infinite-dimensional spaces, there are indeed some density functions $f(x) \in L^1$ with infinite $\ell^2$ norm.

**Non-Example 1** (Square-Unintegrable PDFs). *Let the distribution $F_X$ of a random variable $X$ be*

$$F_X(x) = \begin{cases} 0, & x \in (-\infty, 0) \\ \frac{1}{\sqrt{a}}x^{\frac{1}{2}}, & x \in [0, a] \\ 1, & x \in (a, +\infty) \end{cases}$$

*where $a > 0$ is the parameter; and the density function is then*

$$f(x) = \begin{cases} \frac{1}{2\sqrt{a}}x^{-\frac{1}{2}}, & x \in (0, a) \\ 0, & \text{otherwise} \end{cases}$$

*where $f(x)$ is not square-integrable since $x^{-1}$ increases too fast as $x \to 0$.* ∎

# G    PROOFS OF THEOREMS IN THE MAIN TEXT

## G.1    NOTATIONS

Besides the notations in Section 3.1, a few more are introduced as follows. A unit step function is $W(x - c) = \mathbf{1}_{x \in [c, +\infty)}, c \in \mathbb{R}$. $L^1(\mathcal{X}, \rho)$ denotes the space of absolutely integrable functions.

## G.2    EMBEDDING PROBABILITY MEASURES IN $\mathcal{H}$

PROOF (Theorem 1). Since $F_X(x)$ is absolutely continuous, there exists a density function $f_X(x) \in L^1$ such that

$$\frac{\mathrm{d}}{\mathrm{d}x} F_X(x) = f_X(x) \tag{19}$$

almost everywhere. Since $f_X(x) \in L^2$ , let $M = \|f_X\| < +\infty, \forall g \in \mathcal{H}$, we have

$$\left| \mathbb{E}_X \left[ g(x) \right] \right| = \left| \int_x g(x) \, \mathrm{d}F_X(x) \right| \tag{20}$$

$$= \left| \int_x g(x) f(x) \rho(\mathrm{d}x) \right| \tag{21}$$

$$\leqslant \int_x \left| g(x) \right| \left| f(x) \right| \rho(\mathrm{d}x) \tag{22}$$

$$\leqslant M \|g\|_{\mathcal{H}}, \tag{23}$$

where (23) follows from the Cauchy-Schwarz inequality (Rudin, 1976, Theorem 11.35). Thus, the linear functional $\mathbb{E}_X [\cdot]$ is bounded on $\mathcal{H}$ and

$$\mathbb{E}_X \left[ g(x) \right] = \int_x g(x) \, \mathrm{d}F_X(x) = \int_x f_X(x) g(x) \rho(\mathrm{d}x) = \langle f_X, g \rangle_{\mathcal{H}}, \; \forall g \in \mathcal{H},$$

where $\mu_{\mathrm{p}} := f_X \in \mathcal{H}$ is the embedding of the probability measure in $\mathcal{H}$. Now assume that there exists another element $\mu' \in \mathcal{H}$ such that

$$\mathbb{E}_X \left[ g(x) \right] = \langle \mu', g \rangle_{\mathcal{H}}, \; \forall g \in \mathcal{H}.$$

Since $\mu_p - \mu' \in \mathcal{H}$, we have

$$\begin{aligned} \left\| \mu_p - \mu' \right\|_{\mathcal{H}}^2 &= \langle \mu_p - \mu', \mu_p - \mu' \rangle_{\mathcal{H}} \\ &= \langle \mu_p, \mu_p - \mu' \rangle_{\mathcal{H}} - \langle \mu', \mu_p - \mu' \rangle_{\mathcal{H}} \\ &= \mathbb{E}_X \left[ \mu_p - \mu' \right] - \mathbb{E}_X \left[ \mu_p - \mu' \right] \\ &= 0. \end{aligned}$$

Therefore, the embedding $\mu_p$ for probability measure $p$ in $\mathcal{H}$ is a unique equivalence class of the functions that are equal almost everywhere. ∎

## G.3    INJECTION OF PROBABILITY MEASURE EMBEDDINGS IN $\mathcal{H}$

To prove the injection of probability measure embedding in Theorem 2, a preliminary result of (Dudley, 2002, Lemma 9.3.2) is first introduced.

**Lemma 1.** *If $(\mathcal{X}, \mathrm{d})$ is a metric space, $p$ and $q$ are two probability measures on $\mathcal{X}$, then $\mathbb{E}_{x \sim p(x)} [g] = \mathbb{E}_{x \sim q(x)} [g], \forall g \in C_b(\mathcal{X})$ if and only if $p = q$, where $C_b(\mathcal{X})$ is the space of all bounded continuous functions on $\mathcal{X}$.*

PROOF (Theorem 2). Now we prove this theorem in the following two directions.

**Necessity**. Since the embedding of a probability measure is unique in $\mathcal{H}$, it is easy to see that $\mu_p = \mu_q$ if $p = q$.

**Sufficiency**. Note that, by Weierstrass extreme value theorem (Rudin, 1976, Theorem 4.16), any real continuous function $g \in C(\mathcal{X})$ on the compact space $\mathcal{X}$ is bounded, i.e., $\forall g \in C(\mathcal{X}), \exists M \in \mathbb{R}$ such that $\left| g(x) \right| < M, \forall x \in \mathcal{X}$. It follows that $C(\mathcal{X}) \subset L^2(\mathcal{X})$ since

$$\int_{\mathcal{X}} \left| g(x) \right|^2 \rho(\mathrm{d}x) \leqslant M^2 |\mathcal{X}| < +\infty.$$

Now if $\mu_p = \mu_q$ almost everywhere, we have

$$\left| \mathbb{E}_p\left[g(x)\right] - \mathbb{E}_q\left[g(x)\right] \right| = \left| \langle \mu_p, g \rangle - \langle \mu_q, g \rangle \right| = \left| \langle \mu_p - \mu_q, g \rangle \right| \tag{24}$$

$$\leqslant \left\| \mu_p - \mu_q \right\|_{\mathcal{H}} \|g\|_{\mathcal{H}} = 0, \ \forall g \in C(\mathcal{X}). \tag{25}$$

Thus $p = q$ by Lemma 1. ∎

### G.4   Approximation of Expectation for Discrete/Mixed Distributions in $\mathcal{H}$

While Theorem 1 in the main text only addresses the continuous distributions, a similar result can be found given point-wise continuity conditions for general distributions, which can be decomposed into absolutely continuous and discrete parts (Chung, 2000).

**Theorem 4** (Approximation of Mixed Distribution). *Let $F_{ac}(x)$ be an absolutely continuous distribution function with density function $f_X(x)$; $F_d(x) = \sum_i b_i W(x - a_i)$ a discrete distribution function of point mass at a countable set $\{a_i\}$ such that $b_i > 0$ and $\sum_i b_i = 1$; $F_X(x) = \lambda F_{ac}(x) + (1 - \lambda) F_d(x)$ a mixed distribution function with $\lambda \in (0, 1)$ as the convex combination coefficient. If $f_X(x)$ is square-integrable, and $g(x) \in L^2$ is uniformly continuous at $\{a_i\}$, then there exists a sequence of $\{\mu_{p,n}\} \subset \mathcal{H}$ such that*

$$\lim_{n \to \infty} \langle \mu_{p,n}, g \rangle_{\mathcal{H}} = \mathbb{E}_X\left[g(x)\right]. \tag{26}$$

We start by considering a simple discrete case by the following lemma.

**Lemma 2.** *Let $F_X(x) = W(x - a)$ be a discrete distribution function with point mass at $a \in \mathcal{X}$. If $g(x) \in L^2$ is continuous at $a$, then there exists a sequence of $\{\mu_{p,n}\} \subset \mathcal{H}$ such that*

$$\lim_{n \to \infty} \langle \mu_{p,n}, g \rangle_{\mathcal{H}} = \mathbb{E}_X\left[g(x)\right]. \tag{27}$$

PROOF (Lemma 2). $\forall \varepsilon > 0$, since $g(x)$ is continuous at $a$, there exists a a radius $r > 0$ such that

$$g(a) - \varepsilon \leqslant g(x) \leqslant g(a) + \varepsilon, \ \forall x \in B(a, r)$$

with a positive measure $V = \rho(B(a, r)) > 0$, where $B(a, r) \subset \mathcal{X}$ is a neighbourhood of $r$ around $a$. Define

$$h_\varepsilon(x) = \frac{1}{V} \mathbf{1}_{x \in B(a, r)} \in \mathcal{H}.$$

We have

$$g(a) - \varepsilon < \langle h_\varepsilon, g \rangle_{\mathcal{H}} < g(a) + \varepsilon.$$

Thus,

$$\lim_{n \to \infty} \left\langle h_{\frac{1}{n}}, g \right\rangle_{\mathcal{H}} = g(a) = \mathbb{E}_X\left[g(x)\right].^2$$

∎

Lemma 2 implies that the expected value of a function continuous at the point mass of a delta distribution can be approximated by an inner product in $\mathcal{H}$ with any *arbitrary precision*.

PROOF (Theorem 3). Note that

$$\mathbb{E}_X\left[g(x)\right] = \lambda \int_x g(x) \, \mathrm{d}F_{ac}(x) + (1 - \lambda) \sum_i b_i g(a_i).$$

Since $F_{ac}(x)$ is absolutely continuous, by Theorem 1, there exists a $\mu \in \mathcal{H}$ such that

$$\int_x g(x) \, \mathrm{d}F_{ac}(x) = \langle \mu, g \rangle_{\mathcal{H}}, \ \forall h \in \mathcal{H}. \tag{28}$$

On the other hand, $\forall \varepsilon > 0$, since $g(x)$ is *uniformly continuous* at $\{a_i\}$, there exists a radius $r > 0$ such that $\forall i$,

$$g(a_i) - \varepsilon < g(x) < g(a_i) + \varepsilon, \ \forall x \in B(a_i, r)$$

---

$^2\{h_{\frac{1}{n}}\}$ itself, however, is not a Cauchy sequence, thus it has no limit.

and $V = \rho(B(a_i, r)) > 0$ (translation invariance of Lebesgue measures in $\mathbb{R}^d$). Define

$$h_\varepsilon(x) = \frac{1}{V} \sum_i b_i \mathbf{1}_{x \in B(a_i, r)} \in \mathcal{H}.$$

We have

$$\sum_i b_i g(a_i) - \varepsilon = \sum_i b_i g(a_i) - \varepsilon \sum_i b_i < \langle h_\varepsilon, g \rangle_{\mathcal{H}} < g(a) + \varepsilon \sum_i b_i = \sum_i b_i g(a_i) + \varepsilon,$$

Thus

$$\lim_{n \to \infty} \left\langle h_{\frac{1}{n}}, g \right\rangle_{\mathcal{H}} = \sum_i b_i g(a_i). \tag{29}$$

Combining (28) and (29) leads to

$$\lim_{n \to \infty} \left\langle \lambda \mu + (1 - \lambda) h_{\frac{1}{n}}, g \right\rangle_{\mathcal{H}} = \lambda \int_x g(x) \, \mathrm{d}F_{ac}(x) + (1 - \lambda) \sum_i b_i g(a_i) = \mathbb{E}_X \left[ g(x) \right]. $$

$\blacksquare$

### G.5 Uniform Convergence Rate of Expected Utility Estimators

PROOF (Theorem 3). Assume that $\{X_i\}_{i=1}^n$ and independent and $X_i \in [a_i, b_i]$ almost surely. Let $\bar{X} = \frac{1}{n} \sum_i X_i$.

Per Hoeffding's inequality (Hoeffding, 1963, Theorem 2), for any $\varepsilon > 0$,

$$\Pr\left( \bar{X} - \mathbb{E}\left[ \bar{X} \right] > \varepsilon \right) < \exp\left\{ -\frac{2n^2 \varepsilon^2}{\sum_{i=1}^n (b_i - a_i)^2} \right\}. \tag{30}$$

By symmetry, it also holds true that, for any $\varepsilon > 0$,

$$\Pr\left( \bar{X} - \mathbb{E}\left[ \bar{X} \right] < -\varepsilon \right) < \exp\left\{ -\frac{2n^2 \varepsilon^2}{\sum_{i=1}^n (b_i - a_i)^2} \right\}. \tag{31}$$

Combining one-side inequalities of (30) and (31) leads to

$$\Pr\left( \left| \bar{X} - \mathbb{E}\left[ \bar{X} \right] \right| > \varepsilon \right) < 2\exp\left\{ -\frac{2n^2 \varepsilon^2}{\sum_{i=1}^n (b_i - a_i)^2} \right\} \leqslant 2\exp\left( -\frac{n\varepsilon^2}{2M^2} \right), \ \forall \varepsilon > 0, \tag{32}$$

where $M = \sup(\{|a_1|, \cdots, |a_n|, |b_1|, \cdots, |b_n|\})$.

On the other hand, Bernstein inequality (Bernstein, 1946) also provides an improved revision of Chebyshev's inequality by incorporating both almost-sure bound and variance bound:

$$\Pr\left( \left| \bar{X} - \mathbb{E}\left[ \bar{X} \right] \right| > \varepsilon \right) < 2\exp\left\{ -\frac{n\varepsilon^2}{2\mathrm{Var}(\bar{X}) + 2M\varepsilon/3} \right\}, \ \forall \varepsilon > 0. \tag{33}$$

The proof is completed by setting $X_i = U(S_i, a)$ and taking the lowest bound of (32) and (33) for the tail probability of $\left| \bar{X} - \mathbb{E}\left[ \bar{X} \right] \right|$. $\blacksquare$

