# OpenReview forum: "Transcendental Idealism of Planner: Evaluating Perception from Planning Perspective for Autonomous Driving"
_ICLR.cc/2023/Conference — Submitted to ICLR 2023_

### Official Review · Reviewer_dqEU · 2022-10-15

**Confidence:** 2
**Correctness:** 3
**Technical Novelty And Significance:** 3
**Empirical Novelty And Significance:** 3
**Recommendation:** 6

**Clarity, Quality, Novelty And Reproducibility:**

- The presentation is clear.

- It would be useful if the authors could provide the source code for their method.


**Strength And Weaknesses:**

--- Strengths

- Evaluating the end-to-end perception performance in terms how it affects AV planning has significant practical value. The proposed method is shown to better capture the severity of perception errors compared to the existing methods.

- The paper also provides theoretical reasoning for their proposed method in addition to the empirical results.


--- Weaknesses

- It is not clear whether this metric is generic to different planner architectures.

**Summary Of The Paper:**

This paper presents a method, called transcendental idealism of planner (TIP), to evaluate the end-to-end performance of a perception model in terms of how it affects AV planning. The authors formulate the planning of an autonomous vehicle as an expected utility maximization (EUM) problem. They further show that the objective function can be represented as an inner product between the world state description and the utility function in a Hilbert space, from which they derive the TIP metric.

The authors compared their proposed TIP metric against the traditional perception metrics, including NDS and PKL, on both synthetic and real datasets. They had the human labelers label the scenarios where the two metrics disagree, and the result shows that their TIP metric better captures the severity of perception errors.

**Summary Of The Review:**

Evaluating the end-to-end perception performance in terms how it affects AV planning has significant practical value. The proposed method is shown to better capture the severity of perception errors compared to the existing methods.

---

### Official Review · Reviewer_34kZ · 2022-10-22

**Confidence:** 2
**Correctness:** 3
**Technical Novelty And Significance:** 3
**Empirical Novelty And Significance:** 3
**Recommendation:** 6

**Clarity, Quality, Novelty And Reproducibility:**

- The writing is fluent. The contents are relatively difficult to follow for people unfamiliar with related works.
- There is no released code, but enough implementation and design details are provided.

###### Misc.
- Most figures (especially Fig.1 and Fig. 2) are not black&white-friendly.
- The footnote on page 5 should be on page 4.
- On page 2, for the sentence "or even negative in many common cases", Fig. 1 should show at least one example with negative correlations.
- While Section 2 claims the proposed evaluation only works for model-based planning methods rather than data-driven ones, it would be more accurate to say it works for module-based planning rather than end-to-end ones.

**Strength And Weaknesses:**

###### Pros
- The topic of evaluating perceptions from the planning perspective is new and in great demand.
- The proposed method is well theoretically supported, and its implementation is not complicated.
- Adequate comparisons and case studies to other evaluation methods are provided, which fully demonstrate the proposed method's merits.

###### Cons
- It seems that the definition of the utility function is critical for this evaluation. The authors could discuss more how the choice and the computation of utility functions affect the usage of this evaluation method.
- This method assumes the action set is finite, which may not be satisfied by all planners.
- The definition in Eq. (13) takes the maximum reduction considering all actions. This is the worst-case analysis and is essential for AV, but would it be too conservative (so that the evaluation results may be too far from that of the final chosen action) in practice?

**Summary Of The Paper:**

This paper proposes a new method to evaluate the perception results in autonomous vehicles based on its influences on the following planning module. With theoretical guarantees, the impact of the perception error can be decomposed into two orthogonal complements that are planning-critical and planning-invariant. Experiments on both synthetic and real datasets demonstrate the efficacy of the proposed evaluation method.



----
_Post-rebuttal_:
Thanks for the response.
It is glad to see the detail of Q1 and the updated contents in RS.
For Q2 and Q3, the authors have provided more explanations and discussions while admitting what is (and is not) included in the current work, which is OK to the reviewer.
Therefore, the reviewer keeps the original positive score (6).

**Summary Of The Review:**

Overall this is a solid paper on an interesting research topic with thorough discussions.
The reviewer is not familiar with all aspects and may skip some details.

---

### Official Review · Reviewer_NfXU · 2022-10-24

**Confidence:** 4
**Correctness:** 4
**Technical Novelty And Significance:** 3
**Empirical Novelty And Significance:** Not applicable
**Recommendation:** 5

**Clarity, Quality, Novelty And Reproducibility:**

The paper is clearly explained and the writing quality is good.
The proposed method is novel in formulating the planner optimization problem as a Hilbert inner product between the world state and the utility function.
Reproducibility is hard for the synthetic dataset curated by the authors.


**Strength And Weaknesses:**

Strength:
1. The paper is well written and the approach is well motivated.
2. Results show that the proposed method has higher sensitivity in detecting perception noise, compared to PKL and NDS.

Weakness:
1. It is unclear to me how different methods’ scores are calibrated. Therefore, the proposed method might be just analyzed in the scale it is designed for. Moreover, should we calibrate the same scoring system across different evaluation scenarios? For the proposed TIP, what does a score evaluated from scenario A indicate when compared to a lower score evaluated from scenario B?
2. The proposed TIP is for evaluating perception quality but the paper only shows results of added perception noise to one particular detector. I would like to see how TIP evaluates and compares different object detectors.
3. As shown in Figure 4, the proposed TIP does not yield a monotonic score with respect to the distance. Will this cause confusion in practice?
4. The paper fails to compare with SDE (Deng et al) quantitatively.


**Summary Of The Paper:**

This paper proposes a method for evaluating the autonomy perception module from the perspective of a planner. This is an important topic and starts to receive increasing attention in recent years, yet still underexplored. Specifically, the paper proposes to formulate the expected utility maximization objective as an Hilbert inner product between the world state and the utility function.

The proposed approach is evaluated using synthetic and real data. Results show that the proposed method demonstrates higher sensitivity to perception noise, compared to PKL and NDS.


**Summary Of The Review:**

I am on the fence and would like to see authors' feedback to make the final decision.

---

### Decision · Program_Chairs · 2023-01-20

**Decision:**

Reject

**Justification For Why Not Higher Score:**

As noted above, the paper considers an important, yet under-explored problem. However, there are limitations that need to be addressed.

**Justification For Why Not Lower Score:**

N/A

**Metareview: Summary, Strengths And Weaknesses:**

The paper considers the important problem of evaluating the effectiveness of an autonomous vehicle's perception stack in the context of its influence on the downstream planning module. The paper proposes Transcendental Idealism of Planner (TIP) as a means of quantifying this performance when the planning problem is formulated as one of expected utility maximization. The TIP metric follows from a formulation of the objective function in terms of a Hilbert space inner product between the utility function and the world state. Comparisons against existing metrics on both simulated and real-world data reveal that the proposed TIP metric is more capable of identifying the extent of the perception errors.

As all three reviewers emphasize, the paper considers an important, yet under-explored problem. While a great deal of attention has been paid to evaluating perception algorithms in isolation, the fact that they are increasingly being used to inform downstream planners (e.g., for self-driving cars and other robots) necessitates the existence of a more holistic evaluation. The paper does a nice job conveying the significance of this problem, and describing the TIP metric and its underlying theoretical principles. The reviewers raised a few questions/issues with the initial submission, which the authors made a concerted effort to address through the addition of new experimental results and detailed discussions. However, the method's dependency on the specific nature of the utility function remains unclear as does the extent to which the framework extends to other planning architectures. This is a promising line of work and an updated version of the paper that addresses these remaining concerns would provide a solid contribution to the broader community.

**Summary Of Ac-Reviewer Meeting:**

N/A